# Measuring the Fate of Compost-Derived Phosphorus in Native Soil below Urban Gardens

**DOI:** 10.3390/ijerph16203998

**Published:** 2019-10-19

**Authors:** Gaston E. Small, Sara Osborne, Paliza Shrestha, Adam Kay

**Affiliations:** 1Biology Department, University of St. Thomas, Saint Paul, MN 55105, USA; Osbo0030@gmail.com (S.O.); adkay@stthomas.edu (A.K.); 2Department of Ecology, Evolution, and Behavior, University of Minnesota, Saint Paul, MN 55108, USA; shresthp@umn.edu

**Keywords:** urban agriculture, nutrient, leachate, phosphorus, soil, garden

## Abstract

The heavy reliance on compost inputs in urban gardening provides opportunities to recycle nutrients from the urban waste stream, but also creates potential for buildup and loss of soil phosphorus (P). We previously documented P in leachate from raised-bed garden plots in which compost had been applied, but the fate of this P is not known. Here, we measured P concentrations in soils below four or six-year-old urban garden plots that were established for research. We hypothesize that the soil P concentration and depth of P penetration will increase over time after gardens are established. Soil cores were collected in five garden plots of each age and quantified for inorganic weakly exchangeable P. Inorganic weakly exchangeable P was significantly elevated in native soil below garden plots (>35 cm deep) relative to reference soil profiles, and excess P decreased with increasing depth, although differences between garden plots of different ages were not significant. Our analysis shows that excess P from compost accumulates in native soil below urban garden plots. While urban agriculture has the potential to recycle P in urban ecosystems, over-application of compost has the potential to contribute to soil and water pollution.

## 1. Introduction

Cities represent nodes of resource consumption and waste production in an increasingly urbanized world [1]. Organics waste recycling through composting converts waste products into resources, and has potential to create more circular nutrient flows in urban ecosystems [2]. Many cities have ambitious goals for expanding organic waste recycling programs [3,4,5,6] in order to reduce the waste stream going to landfills. These efforts will also generate more compost, which may pose a dilemma as to how and where to apply this material. 

Urban agriculture (UA) provides one beneficial use for compost, with the promise of recycling nutrients from urban food waste back into the human food system. Outdoor, soil-based UA (including backyard gardening) typically relies on compost application for soil fertility [4] and has expanded rapidly in North America in recent years [7]. However, manure and other organic wastes commonly have low nitrogen:phosphorus (N:P) ratios relative to requirements for crop production, meaning that compost applied based on crop N-demand results in overapplication of P [8,9]. Moreover, many small-scale gardeners may apply compost and other soil amendments at rates far in excess of crops’ ability to assimilate these nutrients [10]. For example, in the Montreal UA system, 27 times more P is applied as compost than is harvested as crops each year, resulting in a P loss or soil build-up of 175 kg P ha^−1^ y^−1^ across the 18 km^2^ of cultivated land in the city [4]. An analysis of urban and peri-urban gardens in three West African cities found large annual surpluses of P (83–780 kg P ha^−1^) resulting in groundwater pollution [11]. In home gardens in the Twin Cities Metropolitan Area (Minneapolis and Saint Paul, MN, USA), gardeners applied an average of 40 times more phosphorus (P) as compost compared to P removed as harvested crops, constituting one of the largest P fluxes on the urban landscape and leading to P buildup in garden soil [12].

Large mass transfers of nutrients into a relatively small area of agricultural production can contribute to the buildup and subsequent export of these nutrients, as has been well-documented in agricultural areas with intensive livestock production, where manure is applied for fertilization [13]. Soils that repeatedly receive excess P inputs can contribute to water pollution for decades to centuries [13]. Legacy P transmission from soils can occur not only from surface runoff, but also from leachate, especially where soil P levels are high [14,15,16]. Although many soils are effective at retaining P at moderate concentrations, high rates of P application can overwhelm this capacity and result in P loss. One long-term agricultural P addition experiment found that soils with less than 60 PPM Olsen P strongly retained P, but P losses in drainage water increased with increasing soil P above this threshold [17].

We have previously documented excessively high concentrations of plant-available P in urban garden soils in the Twin Cities (median Bray P value of 80 PPM) with concentrations increasing with garden age [12], and rates of P loss through leachate in urban gardens that are similar in magnitude to rates of P uptake by crops [18]. However, we know little about the ultimate fate of P lost from urban gardens in the form of leachate. UA is characterized by higher P application rates per unit area and lower P use efficiency compared to conventional rural crop production [12], so the loss of P from garden soils (resulting in a buildup of P in native soil below urban gardens) may be accelerated compared to what has been documented in other agricultural systems [13,19]. In this study, we measured soil P profiles below experimental urban garden plots of two different ages that received annual inputs of compost, along with reference soil profiles below unfertilized turfgrass. We hypothesize that native soil below garden plots is enriched in P relative to reference plots. Furthermore, we predict that soil P concentration and depth of excess P penetration will increase over time after gardens are established.

## 2. Materials and Methods 

This study was conducted in the research garden on the campus of the University of St. Thomas, Saint Paul, Minnesota, USA. Thirty-two 4 m^2^ raised bed garden plots were established in 2011. From 2011–2016, individual plots had received annual inputs of 0–6 kg/m^2^ of composted cow manure, or 0–9 kg/m^2^ of municipal compost (a mixture of yard waste, food waste, and other municipal organics waste) for six years (Table 1). These compost inputs were associated with single-season research projects, and after each growing season, soil from all of the raised-bed garden plots was homogenized and redistributed. As a result, the long-term annual P input to all garden plots was approximately 15 g P/m^2^, slightly lower than median compost-derived P inputs documented for urban gardens in the Twin Cities [12]. The campus garden also includes additional beds for crop production that were established in 2013. These beds have received annual applications of approximately 9 kg/m^2^ of municipal compost (approximately 15 g P/m^2^). These garden soils are characterized as loamy with a high organic matter content (mean 9.4%; loss on ignition method) and circumneutral pH (mean 7.1).

In June to July 2017, soil was sampled in duplicates below 5 of the raised-bed garden plots established in 2011 (referred to hereafter as 6-year old gardens), and 5 locations in the additional crop beds established in 2013 (referred to hereafter as 4-year old gardens). Holes were dug using post-hole diggers, and soil samples were collected at 10 cm intervals up to a depth of 1 m. For each garden soil profile, we collected a reference soil profile below turfgrass located at least 1 m from the edge of the raised-bed garden. Turfgrass has been unfertilized since the establishment of the garden plots. Soil bulk densities (g cm^−3^) were determined by weighing a known volume of soil after oven drying for 48 hours at 50 °C.

One 6-year garden soil profile and accompanying reference soil profile were analyzed for ten different chemically defined P concentrations, using methods modified from [19] and [20]. Five different extractants were used to determine different chemically defined fractions of P: dissolved P (deionized water), weakly exchangeable P (1 M KCl), plant-available P (0.5 M NaHCO_3_), iron-bound P (0.1 M NaOH), and calcium- and magnesium-bound P (0.5 M HCl). These fractions range from highest to lowest bioavailability respectively [21]. For each fraction, inorganic P (P_i_) and total P (P_t_) were determined. Briefly, 20 mL of deionized water, 1 M KCl, 0.5 M NaHCO_3_, 0.1 M NaOH, or 0.5 M HCl was added to 1 g of soil in 50 mL centrifuge tubes. Samples were placed on a horizontal shaker for 24 hours, at 180 oscillations/minute. Samples were then centrifuged, decanted, and split into two equal parts, one for inorganic P and one total P. The sample for total P analysis received 5 mL of 5% Na_2_S_2_O_8_, and was autoclaved on a 35-minute liquid cycle. Both sets of samples were diluted 10:1 with deionized water, and analyzed for phosphate using an HI 96713 Phosphate Low Range Ion Selective Meter (Hannah Instruments, Woonsocket, RI, USA). Duplicate soil samples from each depth were analyzed and average values were used in all calculations.

Based on the results of the analyses on the initial soil profiles, the full set of five 6-year garden soil and reference profiles and five 4-year garden soil and reference profiles (a total of 200 samples, run in duplicate), was analyzed for KCl-P_i_ and KCl-P_t_ as described above. Excess P was calculated for each sample as the difference between the garden soil column P concentration and the reference soil column P concentration from the same depth. We used two-way analysis of variance to test for differences in excess KCl-P_i_ and KCl-P_t_ in native soil underlying garden plots (at depths 40–100 cm) as a function of garden age, soil depth, and age × depth interaction, using JMP Pro 14. Both garden age (4-year and 6-year) and soil depth were treated as a categorical variables, after determining that the relationship between soil depth and excess P was non-linear. The response variable, excess KCl-P_i_, was log_10_(x + A) transformed, where A equals the value required to make the largest negative value equal to 1. The assumption of normal distribution was confirmed by examination of the normal quantile plot of residuals and Shapiro-Wilk test. Inspection of the plot of residuals vs. predicted values confirmed the assumption of variance equality. 

We also calculated excess KCl-P_i_ on a volumetric basis using measured soil bulk density, and we used these values to estimate total excess KCl-P_i_ in the top 70 cm of native soil below garden plots of each age.

## 3. Results

For the garden soil profile in which the full suite of P-extractions was done, relative to the reference soil profile, garden soil in the top 30 cm was most enriched in KCl-P_i_ (6.4 times greater) and KCl-P_t_ (27 times greater) (Table A1). Smaller relative increases were observed for DI-P_i_ (4.3 times greater), DI-P_t_ (1.4 times greater), NaHCO_3_-P_i_ (2.5 times greater), and NaHCO_3_-P_t_ (1.8 times greater). Garden soil was not enriched in NaOH-P_i_, NaOH-P_t_, HCl-P_i_, or HCl-P_t_. Native soil below the garden plot (40–100 cm total depth) was slightly enriched in KCl-P_t_ (five times greater) and KCl-P_i_ (1.5 times greater) relative to the reference profile. No P-enrichment was observed in native soil for other extractions (Table A1).

Across all soil profiles, native soil below garden plots had mean KCl-P_i_ 8 ppm higher compared to corresponding depths from adjacent turfgrass soil cores (Figure 1a). Excess KCl-P_i_ in native soil below gardens was significantly higher at 40 cm depth (near the top of the native soil column) compared to other depths. The difference in excess KCl-P_i_ between 4-year and 6-year garden plots was not significant (P = 0.15), and no significant interactions were observed (Table 2). Native soil below garden plots had mean KCl-P_t_ 9 ppm higher compared to corresponding depths from adjacent turfgrass soil cores (Figure 1b). Neither soil depth, garden age, nor depth × age interaction were significant predictors of KCl-P_t_.

The 6-year garden plots had a total of 18.2 g excess KCl-P_i_ per m^2^, compared to 16.2 g excess KCl-P_i_ per m^2^ for the 4-year garden plots. Native soil from 30–100 cm below the 6-year garden plots had an average of 7.5 g excess KCl-P_i_ per m^2^, compared to 4.7 g excess KCl-P_i_ per m^2^ for the 4-year gardens (Table 3). Dividing by the age of these gardens, the rate of KCl-P_i_ build-up is 1.3 g/m^2^/y for the 6-year old gardens, and 1.2 g/m^2^/y for the 4-year gardens.

## 4. Discussion

Our analysis shows that excess P is present in native soil below urban garden plots, supporting our first hypothesis. Although we did not observe significant differences between native soil P underlying garden plots established 4 and 6 years ago (i.e., lack of direct support for our second hypothesis), the higher KCl-P_i_ in native soil below gardens relative to reference profiles, and the higher excess KCl-P_i_ observed in native soil closer to the garden soil, indicate excess P from compost-amended garden soil that is gradually building up in underlying native soil. Furthermore, the rate of KCl-P_i_ buildup in native soil below garden plots (~1.2 g/m^2^/y) is consistent with rates of P leachate (0.6–1.5 g P/m^2^/y) measured from raised bed gardens to which manure-based compost had been applied [18]. The rate of P buildup in native soil documented here is small relative to the average annual P surplus (the difference between P applied as compost and P removed as harvested crops; approximately 26 g/m^2^/y) of home gardeners in the Twin Cities Metropolitan Area in Minnesota, consistent with previous studies showing that P accumulation in garden soils is the dominant fate of this excess P [12,18]. Nevertheless, our results are the first to show evidence of legacy P gradually accumulating along hydrologic flowpaths in UA systems, similar to what has previously been demonstrated in conventional agriculture systems [13,19].

The relatively low KCl-P_i_ observed in native soil under gardens suggests that this soil has capacity to retain most leached garden P for some years, but over time, storage of this legacy P will increase and ultimately come into contact with groundwater and/or nutrient-sensitive freshwater ecosystems. The capacity of different soils to retain P varies according to soil class, minerology, and pH [22,23,24]. Once the soil is saturated with P, soils can lose the ability to retain additional P applied from compost, increasing the risk of P loss to leachate [25] which can have negative implications for water quality. P from garden soils can also be washed off by stormwater runoff into the nearest storm drains in urban areas with high amounts of paved surfaces during intense precipitation events, ultimately moving downstream and discharging directly into surface waters, which can have adverse water quality effects. Additional studies that determine rate of leachate P under varying precipitation intensities and duration might be beneficial in understanding the likelihood of movement and transport of P from soils.

While household fertilizer use has received significant scientific and regulatory focus in urban nutrient budgets [26,27], the potential effects of overapplication of compost-derived P have heretofore received little attention. This study adds to the small but growing body of evidence that overapplication of compost can result in hotspots of P pollution. While UA is currently a small component of the total urban land use [4,12,28], UA can constitute a significant component of urban greenspace [7], and the implications of excessive P application would be greatly magnified under scenarios of maximizing outdoor UA production [29,30]. Our results also underscore the conclusions from [4] that the capacity of UA systems to assimilate compost-derived nutrients is much smaller than the city’s capacity to produce compost.

This study is intended as a preliminary examination of potential P enrichment in native soil below urban gardens. Future studies should include sampling native soil below a wider array of urban gardens and farms with various soil types and include a wider range of garden ages and management practices. Measurements repeated over time in the same garden plots would also allow for direct quantification of rates of P build-up over time at different depths. Nevertheless, the results reported here indicate that compost-derived P builds up in native soil below urban garden plots at measurable rates over relatively short timescales.

## 5. Conclusions

The use of compost in urban gardens has environmental benefits including recycling nutrients from the municipal waste stream back into the human food system and reducing reliance on synthetically produced nitrogen and mined phosphorus. However, the perception of sustainability surrounding compost use by many urban gardeners may be misleading, as compost-derived P is often applied far in excess of crop nutrient demand, leading to elevated P concentrations in garden soils and loss of excess P from runoff and leachate. Here, we show the first evidence of the accumulation of compost-derived P, that has been lost from urban gardens, into the surrounding environment. These results suggest that, unless carefully managed, compost overapplication could cause urban gardens to become hotspots of nutrient pollution. These results underscore the challenges inherent in recycling urban nutrients through compost and urban agriculture, and the importance of targeting compost applications based on garden soil test results.

## Figures and Tables

**Figure 1 ijerph-16-03998-f001:**
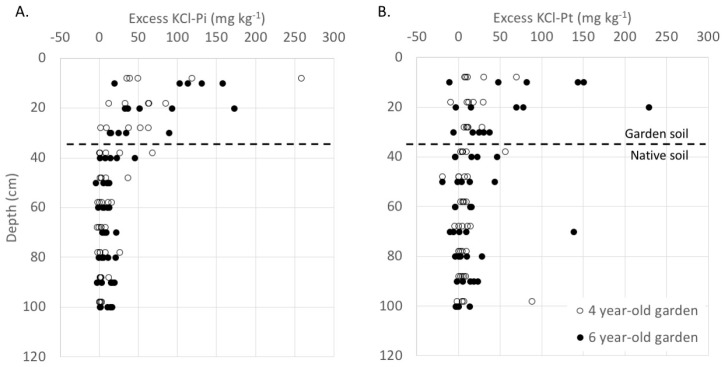
(**a**) Excess KCl-P_i_ (garden plot–reference plot) by depth for soil below garden plots of 4 years (open circles) and 6 years (closed circles). Symbols are slightly offset on Y-axis for legibility. (**b**) Excess KCl-P_t_ (garden plot–reference plot) by depth for soil below garden plots of 4 years and 6 years.

**Table 1 ijerph-16-03998-t001:** Application rates and characteristics of compost used in this study.

Compost Type	Compost Application Rate	Bulk Density	Total Organic Carbon	Total N	Total P	Total K
	(kg dry mass/m^2^/y)	(g dry mass/dm^3^)	%	%	%	%
Manure compost	0–6	120	39.2	1.68	0.60	0.60
Municipal compost	0–9	490	19.2	1.11	0.16	0.41

**Table 2 ijerph-16-03998-t002:** Parameter estimates for log_10_-transformed excess KCl-P_i_ (garden plot–control plot) in the top 70 cm of native soil below raised-bed gardens, as a function of garden age, soil depth, and garden age × soil depth.

Term	Estimate	Std. Error	t Ratio	*P*-Value
Intercept	1.043	0.038	27.6	<0.0001
Garden age (4-year)	−0.055	0.038	−1.45	0.154
Depth (40 cm)	0.210	0.093	2.26	0.028
Depth (50 cm)	−0.022	0.093	−0.23	0.818
Depth (60 cm)	−0.032	0.093	−0.34	0.734
Depth (70 cm)	−0.044	0.093	−0.47	0.642
Depth (80 cm)	−0.023	0.093	−0.25	0.805
Depth (90 cm)	−0.024	0.093	−0.26	0.798
Garden age (4-year) × Depth (40)	0.032	0.093	0.34	0.733
Garden age (4-year) × Depth (50)	0.132	0.093	1.41	0.163
Garden age (4-year) × Depth (60)	0.025	0.093	0.26	0.793
Garden age (4-year) × Depth (70)	−0.103	0.093	−1.11	0.273
Garden age (4-year) × Depth (80)	0.007	0.093	0.08	0.939
Garden age (4-year) × Depth (90)	−0.025	0.093	−0.27	0.788

**Table 3 ijerph-16-03998-t003:** Average excess KCl-P_i_ (garden plot–control plot) on volumetric basis by depth layer. Each value represents excess KCl-P_i_ in 1 dm^3^ (10 cm × 10 cm × 10 cm), allowing for values to be vertically integrated. Depths ≤ 30 cm (italicized) were a garden soil/compost mixture; depths > 30 cm is native soil.

Depth (cm)	4 year-old Garden (mg KCl-P_i_/1 dm^3^)	6 year-old Garden (mg KCl-P_i_/1 dm^3^)
*10*	*76.0*	*91.8*
*20*	*48.2*	*55.5*
*30*	*38.2*	*34.4*
40	20.4	20.7
50	9.9	6.5
60	3.6	6.3
70	1.1	8.5
80	8.0	8.6
90	3.9	10.9
100	0.2	13.4

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
