# Peer review of "Measuring the Fate of Compost-Derived Phosphorus in Native Soil below Urban Gardens"

_ijerph, 2019, doi:10.3390/ijerph16203998_

Round 1

Reviewer 1 Report

The manuscript reports results from a study about P leaching from soils devoted to urban agriculture amended with compost. Although it is a very simple work, the subject has received little attention so far, it is well designed and the results are very interesting. In any case, some additional information about the soil and composts used would make the manuscript more complete.

Specific comments:

Line 38 “organic wastes”

Line 70 Please excuse my ignorance, what is the Twin Cities?

Lines 75-76 It is possible to have some composition data of the composts used? And an approximate range of rates added

Line 86 “Turfgrass”

Figure 1 Please use international units for concentration, this is, mg kg-1

Line 113 I think the reader would appreciate one paragraph or a table summarizing the main properties of the soils, pH, texture, OC, etc… as well as, if possible, soil class

Lines 128-129 The excess P stocks correspond to what soil depth? And how were they calculated?

Table 2 Why are P concentrations expressed in mg/L? Also, I would suggest the authors to switch order of the columns, showing first the 4-year soils and then the 6-year soils

Line 170 and following: Soil class and mineralogy can also modify significantly the capacity for P retention and P leaching; I would suggest to elaborate a little further around line 189 about how different soils have different P retention capacity

Table A1 I would suggest moving this table to the main article, that is short, instead of keeping it as an appendix

Author Response

Comments and Suggestions for Authors

The manuscript reports results from a study about P leaching from soils devoted to urban agriculture amended with compost. Although it is a very simple work, the subject has received little attention so far, it is well designed and the results are very interesting. In any case, some additional information about the soil and composts used would make the manuscript more complete.

Response: Thank you for this assessment and the constructive comments.  The manuscript has been strengthened as a result of implementing these suggestions.

Specific comments:

Line 38 “organic wastes”

Response: We have made this correction.

Line 70 Please excuse my ignorance, what is the Twin Cities?

Response: We have added clarification that the Twin Cities Metropolitan Area refers to Minneapolis and Saint Paul, Minnesota, USA (line 47).

Lines 75-76 It is possible to have some composition data of the composts used? And an approximate range of rates added

Response: Yes, we have now added additional details (lines 76-92) and a new Table 1, describing mass and composition of compost added.

Line 86 “Turfgrass”

Response: We have made this correction.

Figure 1 Please use international units for concentration, this is, mg kg-1

Response: We have made this correction.

Line 113 I think the reader would appreciate one paragraph or a table summarizing the main properties of the soils, pH, texture, OC, etc… as well as, if possible, soil class

Response: We have added this information (lines 92-94).

Lines 128-129 The excess P stocks correspond to what soil depth? And how were they calculated?

Response: Excess P was calculated as the difference between measured P in the garden soil column at a given depth and the measured P in the adjacent control (turfgrass) soil column from the equivalent depth.  Excess P stocks were calculated as the sum of excess P in native soil underlying garden plots (from depths 40-100 cm).  See lines 120-128.

Table 2 Why are P concentrations expressed in mg/L? Also, I would suggest the authors to switch order of the columns, showing first the 4-year soils and then the 6-year soils

Response: The objective in this table (now Table 3) was to present calculated excess P volumetrically, to allow for the calculation of excess P in the soil column.  For clarification, I have replaced liters with “1000 cm3”.  I have also added additional explanation in the table legend.  Per your suggestion, I have also switched the order of these two columns.

Line 170 and following: Soil class and mineralogy can also modify significantly the capacity for P retention and P leaching; I would suggest to elaborate a little further around line 189 about how different soils have different P retention capacity

Response: Thank you for this suggestion.  We have added text making this point and added appropriate references (lines 199-201).

Table A1 I would suggest moving this table to the main article, that is short, instead of keeping it as an appendix

Response: I am certainly open to this suggestion; however, I was having problems with formatting, as the table is slightly larger than one page and must be turned sideways.  I will leave it to the discretion of the editors and copy editors as to whether to keep it as an appendix or move it into the main body of the article.

Reviewer 2 Report

The paper reports the results on investigation on accumulation of phosphorous in urban soils fertilized with compost.

GENERAL COMMENTS

The manuscript worth of attention because of the growing interest of compost, as a result of a circular economy process for reuse of organic wastes, but lack on many aspects and need to be revised.

SPECIFIC COMMENTS

L29-30 It is true that composting of organic wastes is process that fits the definition the circularity. Composting has been study since many years for the use of the final product for agricultural purposes (that are similar to those of the Urban Agriculture). The process has some limitations because of the characteristics of the final product that can have some negative effects on plats, especially when the process is not correctly conduced. Furthermore composting is responsible of Green House Gasses emission. These limitations have to be reminded when the process is introduced/described.

Authors find some information about these aspects in the following articles

M.P.Bernal C. Paredes, M.A.Sánchez-Monedero, J.Cegarra 1998 Maturity and stability parameters of composts prepared with a wide range of organic wastes Bioresource Technology. 63 (1), 91-99 Bernal M.P., Alburquerque J. A., Moral ., 2009 Composting of animal manures and chemical criteria for compost maturity assessment. A review. Bioresource Technology.10 (22) 5444-5453 Pampuro N., Bisaglia C., Romano E. et Al.. 2017. Phytotoxicity and chemical characterization of compost derived from pig slurry solid fraction for organic pellet production. Agriculture, 7 (11) Article number 94 Pampuro N., Bertora C, Sacco D., Dinuccio E, Grignani C., Balsari P., Cavallo E., Bernal M.P. 2017. Fertilizer value and GHG emissions of pellets from the solid fraction of pig slurry compost. Journal of Agricultural Science, 155, 1646–1658. Pampuro N., Busato P., Cavallo E.  2018. Gaseous emission after soil application of pellet made from composted pig slurry solid fraction: effect of application method and pellet diameter. Agriculture, 8 (8) Article number 119

L45-50 People out of US not necessarily known what Twin Cities are.

L66 Reference [19] is reported as in relation to agricultural systems. Actually it refers to the Everglades, that in my knowledge is not one of the most common US agricultural system. Furthermore the reference date 1993. I’m sure that the authors can find other scientific papers dealing with pollution from fertilizers, and phosphorus, to compare results and make reference to.

L69-71 The goals of the research are presented as an hypothesis. In discussion and conclusion I did not find any statement regarding these hypothesis. Thus it should be better to introduce the goals of the research as “ This paper investigate the …..” or something similar

L80 The quantity of compost delivered to the plots is referred in height (in cm). It is unusual because. In scientific paper it is reported the quantity (kg/m2 or t/ha) describing the physical and chemical characteristics of the product. This is an important aspect other researchers/practitioners/end user need to compare results and predict the effect of compost distribution in other soils/conditions

L107-111 It is reported JMP software for statistical analysis but it is not described what kind of statistical analysis has been performed and if the assumption required (normal distribution and homoscedasticity of data in case of parametric tests) are satisfied

L113-119 The enrichment in phosphorus is described with a number in bracket. It could not be clear to all of the readers. Explain the notation or adopt a different one. Furthermore it is not clear what the enrichment is compared to (eg:  x6.4 = 6.4 times what?)

Author Response

GENERAL COMMENTS

The manuscript worth of attention because of the growing interest of compost, as a result of a circular economy process for reuse of organic wastes, but lack on many aspects and need to be revised.

Response: We appreciate the specific comments listed below.  We have done our best to address each of them.

SPECIFIC COMMENTS

L29-30 It is true that composting of organic wastes is process that fits the definition the circularity. Composting has been study since many years for the use of the final product for agricultural purposes (that are similar to those of the Urban Agriculture). The process has some limitations because of the characteristics of the final product that can have some negative effects on plats, especially when the process is not correctly conduced. Furthermore composting is responsible of Green House Gasses emission. These limitations have to be reminded when the process is introduced/described.

Response: Although the main focus of our manuscript is on the specific challenge of composting (the issue of P build-up), I do not think it is necessary to discuss other limitations of compost use in the opening sentences.  It is the case that many cities in the U.S. have ambitious goals for expanding composting, and is generally viewed as contributing to sustainability.   While aerobic composting does produce greenhouse gas emissions, this may be less of an impact than methane emissions that could result from this material decomposing in a landfill.  A full environmental life-cycle analysis of compost is outside of the scope of this manuscript.

Authors find some information about these aspects in the following articles

M.P.Bernal C. Paredes, M.A.Sánchez-Monedero, J.Cegarra 1998 Maturity and stability parameters of composts prepared with a wide range of organic wastes Bioresource Technology. 63 (1), 91-99 Bernal M.P., Alburquerque J. A., Moral ., 2009 Composting of animal manures and chemical criteria for compost maturity assessment. A review. Bioresource Technology.10 (22) 5444-5453 Pampuro N., Bisaglia C., Romano E. et Al.. 2017. Phytotoxicity and chemical characterization of compost derived from pig slurry solid fraction for organic pellet production. Agriculture, 7 (11) Article number 94 Pampuro N., Bertora C, Sacco D., Dinuccio E, Grignani C., Balsari P., Cavallo E., Bernal M.P. 2017. Fertilizer value and GHG emissions of pellets from the solid fraction of pig slurry compost. Journal of Agricultural Science, 155, 1646–1658. Pampuro N., Busato P., Cavallo E.  2018. Gaseous emission after soil application of pellet made from composted pig slurry solid fraction: effect of application method and pellet diameter. Agriculture, 8 (8) Article number 119

Response: We appreciate these suggestions.  As described above, we are opting to keep the manuscript narrowly focused on the issue of phosphorus buildup in soils below urban gardens.  These would be excellent reference for a more general discussion of environmental challenges of compost usage. Please note that we have added some additional references in the revised manuscript.

L45-50 People out of US not necessarily known what Twin Cities are.

Response: Thank you.  We have added clarification that the Twin Cities Metropolitan Area consists of the cities of Minneapolis and Saint Paul, Minnesota.

L66 Reference [19] is reported as in relation to agricultural systems. Actually it refers to the Everglades, that in my knowledge is not one of the most common US agricultural system. Furthermore the reference date 1993. I’m sure that the authors can find other scientific papers dealing with pollution from fertilizers, and phosphorus, to compare results and make reference to.

Response: Thank you. We have replaced that reference with a more appropriate, and more recent, reference.

L69-71 The goals of the research are presented as an hypothesis. In discussion and conclusion I did not find any statement regarding these hypothesis. Thus it should be better to introduce the goals of the research as “ This paper investigate the …..” or something similar

Response: Thank you for pointing this out.  We have revised the opening sentences in the Discussion, making reference to the two hypotheses posed at the end of the Introduction.

L80 The quantity of compost delivered to the plots is referred in height (in cm). It is unusual because. In scientific paper it is reported the quantity (kg/m2 or t/ha) describing the physical and chemical characteristics of the product. This is an important aspect other researchers/practitioners/end user need to compare results and predict the effect of compost distribution in other soils/conditions

Response: We have added several sentences in the beginning of the Materials and Methods section (lines 76-94) describing the mass (kg/m2) and characteristics of compost added.  We have also included a table (new Table 1) with this information.

L107-111 It is reported JMP software for statistical analysis but it is not described what kind of statistical analysis has been performed and if the assumption required (normal distribution and homoscedasticity of data in case of parametric tests) are satisfied

Response: We have clarified that we used multiple regression analysis, and that the data conformed to assumptions of normal distribution and homoscedasticity (lines 124-127).

L113-119 The enrichment in phosphorus is described with a number in bracket. It could not be clear to all of the readers. Explain the notation or adopt a different one. Furthermore it is not clear what the enrichment is compared to (eg:  x6.4 = 6.4 times what?)

Response: I have revised that sentence to make it more clear that the numbers in parentheses refer to how many time greater P concentration in the garden soil column exceeds that of the reference soil column).

Round 2

Reviewer 2 Report

The paper has been improved by the authors that have addressed mostly of the comments raised on the first version of the manuscript.

GENERAL COMMENTS

The article approaches the interesting and relevant subjects of use of compost as fertilizer in urban area. I partially agree on the statement of the focus of the manuscript on P for not mentioning the environmental problem that composting and compost application on soil can generate. If the limitations are not reported it seems to the reader not familiar with the subject that composting and compost application to soil are THE SOLUTION for waste disposal when it is not. P and heavy metals (that the compos is reach) can accumulate into the soil, so compost application can assume the form a diluted landfilling. This environmental concerns raises ethical problem in the waste management that the reviewer cannot ignore and have to point out.

SPECIFIC COMMENTS

L87 Table 1 should contain some more physical/chemical characteristics of the compost. Replace L with dm3  (L is adopted for liquid materials)

L117-118 Specify what test/procedure have been adopted to check for normal distribution and homoscedasticity.

Author Response

GENERAL COMMENTS

The article approaches the interesting and relevant subjects of use of compost as fertilizer in urban area. I partially agree on the statement of the focus of the manuscript on P for not mentioning the environmental problem that composting and compost application on soil can generate. If the limitations are not reported it seems to the reader not familiar with the subject that composting and compost application to soil are THE SOLUTION for waste disposal when it is not. P and heavy metals (that the compost is reach) can accumulate into the soil, so compost application can assume the form a diluted landfilling. This environmental concerns raises ethical problem in the waste management that the reviewer cannot ignore and have to point out.

Response: Thank you for providing a second review of our manuscript.  I do very much agree with the Reviewer’s concern that it is important to avoid presenting composting and compost application as a panacea for recycling organics waste.  The focus of our entire manuscript (and indeed, my entire research program) is on P pollution stemming from compost application.  The specific suggestion in the previous round of reviews was to add specific text in the opening paragraph addressing problems with composting.  I note that the last sentence in the opening paragraph does address the dilemma of compost management as more compost is produced in cities (lines 32-33), and the entire introduction after the opening paragraph (lines 34-71) describes the issue of nutrient over-application from compost use.  Likewise, our conclusion criticizes “the perception of sustainability surrounding compost use” (lines 221-222), and warns that compost over-application “could cause urban gardens to become hotspots of nutrient pollution (lines 226-227).  We also believe that issues related to heavy metals, pesticide residue, and greenhouse gas emissions are also important in assessing the overall sustainability of compost use, but these issues are outside of the scope of our study.  I hope to see these issues addressed elsewhere in this special issue, however.

SPECIFIC COMMENTS

L87 Table 1 should contain some more physical/chemical characteristics of the compost. Replace L with dm (L is adopted for liquid materials)

Response: Thank you for this suggestion.  We have added information on compost Total Organic Carbon and Total K to Table 1.  This represents the complete set of physical/chemical characteristics that we have for this compost.  We replaced L with dm3, per the reviewer’s suggestion.

L117-118 Specify what test/procedure have been adopted to check for normal distribution and homoscedasticity.

Response: This comment led us to carefully re-examine the assumptions underlying our statistical model.  Upon closer inspection, we decided that the relationship between soil depth and excess P did not conform to the assumption of linearity, so we now treat soil depth as a categorical variable.  We also used a log transformation for the response variable to ensure that residuals are normally distributed.  We added text describing that the assumption of normal distribution was confirmed by examination of the normal quantile plot of residuals and the Shapiro-Wilk W test, and the assumption of variance equality was confirmed based on inspection of the plot of residuals vs. predicted values (lines 120-124).  The results of this updated analysis are found in lines 139-145 and in Table 2.  It is important to note that this re-analysis does not change our general conclusions in any way.